# Overinterpretation reveals image classification model pathologies

## Abstract

Image classifiers are typically scored on their test set accuracy, but high accuracy can mask a subtle type of model failure. We find that high scoring convolutional neural networks (CNNs) on popular benchmarks exhibit troubling pathologies that allow them to display high accuracy even in the absence of semantically salient features. When a model provides a high-confidence decision without salient supporting input features, we say the classifier has overinterpreted its input, finding too much class-evidence in patterns that appear nonsensical to humans. Here, we demonstrate that neural networks trained on CIFAR-10 and ImageNet suffer from overinterpretation, and we find models on CIFAR-10 make confident predictions even when 95% of input images are masked and humans cannot discern salient features in the remaining pixel-subsets. Although these patterns portend potential model fragility in real-world deployment, they are in fact valid statistical patterns of the benchmark that alone suffice to attain high test accuracy. Unlike adversarial examples, overinterpretation relies upon unmodified image pixels. We find ensembling and input dropout can each help mitigate overinterpretation.

## 1 Introduction

Well-founded decisions by machine learning (ML) systems are critical for high-stakes applications such as autonomous vehicles and medical diagnosis. Pathologies in models and their respective training datasets can result in unintended behavior during deployment if the systems are confronted with novel situations. For example, a medical image classifier for cancer detection attained high accuracy in benchmark test data, but was found to base decisions upon presence of rulers in an image (present when dermatologists already suspected cancer) (Patel, 2017). We define model *overinterpretation* to occur when a classifier finds strong class-evidence in regions of an image that contain no semantically salient features. Overinterpretation is related to overfitting, but overfitting can be diagnosed via reduced test accuracy. Overinterpretation can stem from true statistical signals in the underlying dataset distribution that happen to arise from particular properties of the data source (e.g., dermatologists' rulers). Thus, overinterpretation can be harder to diagnose as it admits decisions that are made by statistically valid criteria, and models that use such criteria can excel at benchmarks. We demonstrate overinterpretation occurs with unmodified subsets of the original images; in contrast to *adversarial examples* that modify images with extra information, overinterpretation is based on real patterns already present in the training data that also generalize to the test distribution.

Hidden statistical signals of benchmark datasets can result in models that overinterpret or do not generalize to new data from a different distribution. Computer vision (CV) research relies on datasets like CIFAR-10 (Krizhevsky, 2009) and ImageNet (Russakovsky et al., 2015) to provide standardized performance benchmarks. Here, we analyze the overinterpretation of popular CNN architectures on these benchmarks to characterize pathologies.

Revealing overinterpretation requires a systematic way to identify which features are used by a model to reach its decision. Feature attribution is addressed by a large number of interpretability methods, although they propose differing explanations for the decisions of a model. One natural explanation for image classification lies in the set of pixels that is sufficient for the model to make a confident prediction, even in the absence of information about the rest of the image. In the example of the medical image classifier for cancer detection, one might identify the pathological behavior by finding pixels depicting the ruler alone suffice for the model to confidently output the same classifications.

This idea of Sufficient Input Subsets (SIS) has been proposed to help humans interpret the decisions of black-box models (Carter et al., 2019). An SIS subset is a minimal subset of features (e.g., pixels) that suffices to yield a class probability above a certain threshold with all other features masked.

We demonstrate that classifiers trained on CIFAR-10 and ImageNet can base their decisions on SIS subsets that contain few pixels and lack human understandable semantic content. Nevertheless, these SIS subsets contain statistical signals that generalize across the benchmark data distribution, and we are able to train classifiers on CIFAR-10 images missing 95% of their pixels and ImageNet images missing 90% of their pixels with minimal loss of test accuracy. Thus, these benchmarks contain inherent statistical shortcuts that classifiers optimized for accuracy can learn to exploit, instead of learning more complex *semantic* relationships between the image pixels and the assigned class label. While recent work suggests adversarially robust models base their predictions on more semantically meaningful features (Ilyas et al., 2019), we find these models suffer from overinterpretation as well. As we subsequently show, overinterpretation is not only a conceptual issue, but can actually harm overall classifier performance in practice. We find model ensembling and input dropout partially mitigate overinterpretation, increasing the semantic content of the resulting SIS subsets. However, this mitigation is not a substitute for better training data, and we find that overinterpretation is a statistical property of common benchmarks. Intriguingly, the number of pixels in the SIS rationale behind a particular classification is often indicative of whether the image will be correctly classified.

It may seem unnatural to use an interpretability method that produces feature attributions that look uninterpretable. However, we do not want to bias extracted rationales towards human visual priors when analyzing a model's pathologies, but rather faithfully report the features used by a model. To our knowledge, this is the first analysis showing one can extract nonsensical features from CIFAR-10 and ImageNet that intuitively should be insufficient or irrelevant for a confident prediction, yet are alone sufficient to train classifiers with minimal loss of performance. Our contributions include:

- We discover the pathology of overinterpretation and find it is a common failure mode of ML models, which latch onto non-salient but statistically valid signals in datasets (Section 4.1).
- We introduce Batched Gradient SIS, a new masking algorithm to scale SIS to high-dimensional inputs and apply it to characterize overinterpretation on ImageNet (Section 3.2).
- We provide a pipeline for detecting overinterpretation by masking over 90% of each image, demonstrating minimal loss of test accuracy, and establish lack of saliency in these patterns through human accuracy evaluations (Sections 3.3, 4.2, 4.3).
- We show misclassifications often rely on smaller and more spurious feature subsets suggesting overinterpretation is a serious practical issue (Section 4.4).
- We identify two strategies for mitigating overinterpretation (Section 4.5). We demonstrate that overinterpretation is caused by spurious statistical signals in training data, and thus training data must be carefully curated to eliminate overinterpretation artifacts.

## 2 RELATED WORK

There has been substantial research on understanding dataset bias in CV (Torralba & Efros, 2011; Tommasi et al., 2017) and the fragility of image classifiers deployed outside of the benchmark setting (Rosenfeld et al., 2018). CNNs in particular have been conjectured to pick up on localized features like texture instead of more global features like object shape (Gatys et al., 2017; Brendel & Bethge, 2019). Other work has shown deep image classifiers can make confident predictions on nonsensical patterns (Nguyen et al., 2015; Ilyas et al., 2019), though these adversarial examples synthesize artificial images or modify real images with auxiliary information. In contrast, we demonstrate overinterpretation of unmodified subsets of actual training images, indicating the patterns are already present in the original dataset. We further demonstrate that such signals in training data actually generalize to the test distribution. Hooker et al. (2019) found sparse pixel subsets suffice to attain high classification accuracy on popular image classification datasets, but evaluate interpretability methods rather than demonstrate spurious features or discover overinterpretation. In natural language processing (NLP), Feng et al. (2018) explored model pathologies using a similar technique, but did not analyze whether the semantically spurious patterns the models rely on are a statistical property of the dataset. Other research has demonstrated the presence of various spurious statistical shortcuts in major NLP benchmarks, showing this problem is not unique to CV (Niven & Kao, 2019).

## 3 METHODS

### 3.1 DATASETS AND MODELS

CIFAR-10 (Krizhevsky, 2009) and ImageNet (Russakovsky et al., 2015) have become two of the most popular image classification benchmarks. Most image classifiers are evaluated by the CV community based on their accuracy in one of these benchmarks. We also use the CIFAR-10-C dataset (Hendrycks & Dietterich, 2019) to evaluate the extent to which our CIFAR-10 models can generalize to out-of-distribution (OOD) data. CIFAR-10-C contains variants of CIFAR-10 test images altered by various corruptions (such as Gaussian noise, motion blur, and snow). Where computing sufficient input subsets on CIFAR-10-C images, we use a uniform random sample of 2000 images across the entire CIFAR-10-C set. We use the ILSVRC2012 ImageNet dataset.

For CIFAR-10, we explore three common CNN architectures: a deep residual network with depth 20 (ResNet20) (He et al., 2016a), a v2 deep residual network with depth 18 (ResNet18) (He et al., 2016b), and VGG16 (Simonyan & Zisserman, 2014). We train these networks using cross-entropy loss optimized via SGD with Nesterov momentum (Sutskever et al., 2013) and employ standard data augmentation strategies (He et al., 2016b) (additional details in Section S1). After training many CIFAR-10 networks individually, we construct four different ensemble classifiers by grouping various networks together. Each ensemble outputs the average prediction over its member networks (specifically, the arithmetic mean of their logits). For each of three architectures, we create a corresponding homogeneous ensemble by individually training five networks of that architecture. Each network has a different random initialization, which suffices to produce substantially different models despite having been trained on the same data (Osband et al., 2016). Our fourth ensemble is heterogeneous, containing all 15 networks (5 replicates of each of 3 distinct CNN architectures).

For ImageNet, we use a pre-trained Inception v3 model (Szegedy et al., 2016) that achieves 22.55% and 6.44% top-1 and top-5 error (Paszke et al., 2019).

### 3.2 DISCOVERING SUFFICIENT FEATURES

**CIFAR-10.** We interpret the feature patterns learned by CIFAR-10 CNNs using the Sufficient Input Subsets (SIS) procedure (Carter et al., 2019), which produces rationales (SIS subsets) of a black-box model's decision-making. SIS subsets are minimal subsets of input features (pixels) whose values alone suffice for the model to make the same decision as on the original input. Let $f_c(x)$ denote the probability that an image $x$ belongs to class $c$. An SIS subset $S$ is a minimal subset of pixels of $x$ such that $f_c(x_S) \geq \tau$, where $\tau$ is a prespecified confidence threshold and $x_S$ is a modified input in which all information about values outside $S$ are masked. We mask pixels by replacement with the mean value over all images (equal to zero when images have been normalized), which is presumably least informative to a trained classifier (Carter et al., 2019). SIS subsets are found via a local backward selection algorithm applied to the function giving the confidence of the predicted (most likely) class.

**ImageNet.** At present, it is computationally infeasible to scale the original SIS backward selection procedure to ImageNet. We introduce Batched Gradient SIS, a gradient-based method to find sufficient input subsets on high-dimensional inputs. Rather than separately masking every remaining pixel at each iteration to find the pixel whose masking least reduces $f$, we use the gradient of $f$ with respect to the input pixels $\mathbf{x}$ and mask $M$, $\nabla_M f(\mathbf{x} \odot (1 - M))$, to order pixels (via a single backward pass). Instead of masking only one pixel per iteration, we mask larger subsets of $k \geq 1$ pixels per iteration. Given $p$ input features, our Batched Gradient FindSIS procedure finds each SIS subset in $\mathcal{O}(\frac{p}{k})$ evaluations of $\nabla f$ (as opposed to $\mathcal{O}(p^2)$ evaluations of $f$ in FindSIS (Carter et al., 2019)). The complete Batched Gradient SIS algorithm is presented in Section S5.

### 3.3 DETECTING OVERINTERPRETATION

We produce sparse variants of all train and test set images retaining 5% (CIFAR-10) or 10% (ImageNet) of pixels in each image. Our goal is to identify sparse pixel-subsets that contain feature patterns the model identifies as strong class-evidence as it classifies an image. We identify pixels to retain based on sorting by SIS BackSelect (Carter et al., 2019) (CIFAR-10) or our Batched Gradient BackSelect procedure (ImageNet). These backward selection (BS) pixel-subset images contain the

final pixels (with their same RGB values as in the original images) while all other pixels' values are replaced with zero. Note that we apply backward selection to the function giving the confidence of the *predicted* class from the original model to prevent adding information about the true class for misclassified images, and we use the true labels for training/evaluating models on pixel-subsets. As backward selection is applied locally on each image, the specific pixels retained differ across images.

We train new classifiers on solely these pixel-subsets of training images and evaluate accuracy on corresponding pixel-subsets of test images to determine whether such pixel-subsets are statistically valid for generalization in the benchmark. We use the same training setup and hyperparameters (Section 3.1) without data augmentation of training images (results with data augmentation in Table S1). We consider a model to overinterpret its input when these signals can generalize to test data but lack semantic meaning (Section 3.4).

### 3.4 HUMAN CLASSIFICATION BENCHMARK

To evaluate whether sparse pixel-subsets of images can be accurately classified by humans, we asked four participants to classify images containing various degrees of masking. We randomly sampled 100 images from the CIFAR-10 test set (10 images per class) that were correctly and confidently ($\geq 99\%$ confidence) classified by our models, and for each image, kept only 5%, 30%, or 50% of pixels as ranked by backward selection (all other pixels masked). Backward selection image subsets are sampled across our three models. Since larger subsets of pixels are by construction supersets of smaller subsets identified by the same model, we presented each batch of 100 images in order of increasing subset size and shuffled the order of images within each batch. Users were asked to classify each of the 300 images as one of the 10 classes in CIFAR-10 and were not provided training images. The same task was given to each user (and is shown in Section S4).

## 4 RESULTS

### 4.1 CNNs CLASSIFY IMAGES USING SPURIOUS FEATURES

**CIFAR-10.** Figure 1 shows example SIS subsets (threshold 0.99) from CIFAR-10 test images (additional examples in Section S2). These SIS subset images are confidently and correctly classified by each model with $\geq 99\%$ confidence toward the predicted class. We observe these SIS subsets are highly sparse and the average SIS size at this threshold is $< 5\%$ of each image (see Figure 5), suggesting these CNNs confidently classify images that appear nonsensical to humans (Section 4.3), leading to concern about their robustness and generalizability.

We retain 5% of pixels in each image using local backward selection and mask the remaining 95% with zeros (Section 3.3) and find models trained on full images classify these pixel-subsets as accurately as full images (Table 1). Figure 2a shows the pixel locations and confidence of these 5% pixel-subsets across all CIFAR-10 test images. Moreover, the CNNs are more confident on these pixels subsets than on full images: the mean drop in confidence for the predicted class between original images and these 5% subsets is $-0.035$ (std dev. $= 0.107$), $-0.016$ (0.094), and $-0.012$ (0.074) computed over all CIFAR-10 test images for our ResNet20, ResNet18, and VGG16 models, respectively, suggesting severe overinterpretation (negative values imply greater confidence on the 5% subsets). We find pixel-subsets chosen via backward selection are significantly more predictive than equally large pixel-subsets chosen uniformly at random from each image (Table 1).

We also find SIS subsets confidently classified by one model do not transfer to other models. For instance, 5% pixel-subsets derived from CIFAR-10 test images using one ResNet18 model (which classifies them with $94.8\%$ accuracy) are only classified with $25.8\%$, $29.2\%$, and $27.5\%$ accuracy by another ResNet18 replicate, ResNet20, and VGG16 models, respectively, suggesting there exist many different statistical patterns that a flexible model might learn to rely on, and thus CIFAR-10 image classification remains a highly underdetermined problem. Producing high-capacity classifiers that make predictions for the right reasons may require clever regularization strategies and architecture design to ensure models favor salient features over such sparse pixel subsets.

While recent work has suggested semantics can be better captured by models that are robust to adversarial inputs that fool standard neural networks via human-imperceptible modifications to images (Madry et al., 2017; Santurkar et al., 2019), we explore a wide residual network that is

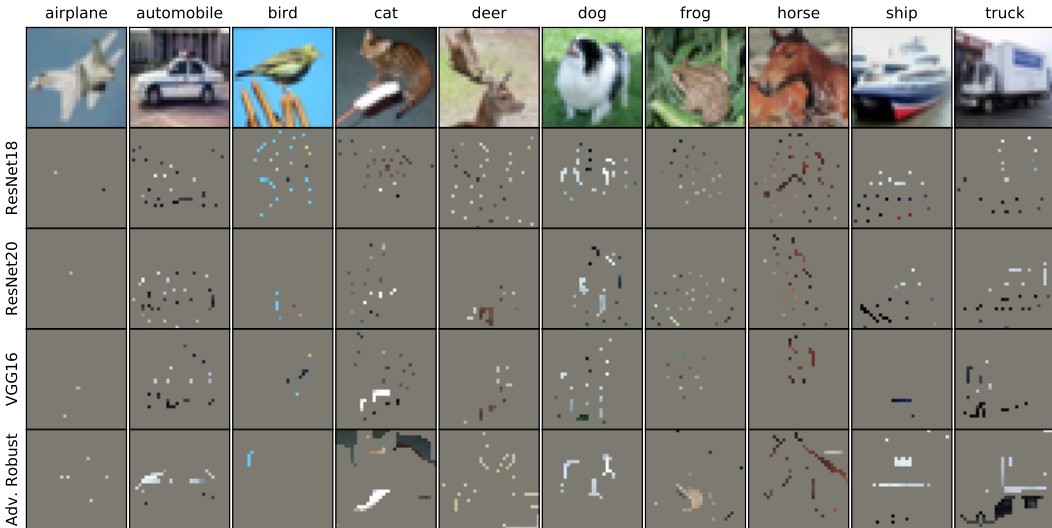

Figure 1: Sufficient input subsets (SIS) for a sample of CIFAR-10 test images (top). Each SIS image shown below is classified by the respective model with $\geq 99\%$ confidence. The "Adv. Robust" (pre-trained adversarially robust) model we use is from Madry et al. (2017) and robust to $l_\infty$ perturbations.

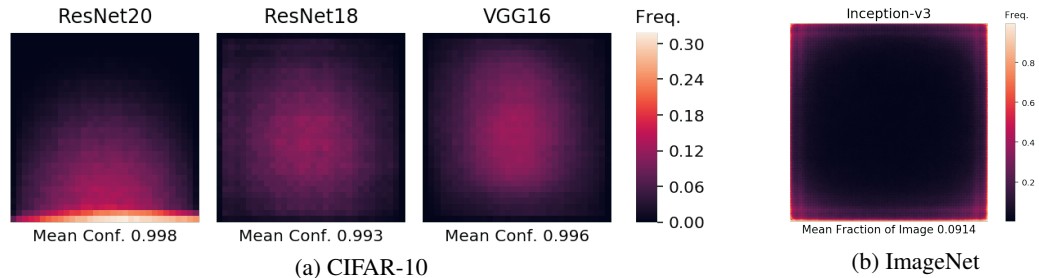

Figure 2: Heatmaps of pixel locations comprising pixel-subsets. Frequency indicates fraction of subsets containing each pixel. **(a)** 5% pixel-subsets across CIFAR-10 test set for each model. Mean confidence indicates confidence on 5% pixel-subsets. **(b)** Sufficient input subsets (threshold 0.9) across ImageNet validation images from Inception v3.

adversarially robust for CIFAR-10 classification (Madry et al., 2017) and find evidence of overinterpretation (Figure 1). This finding suggests adversarial robustness alone does not prevent models from overinterpreting spurious signals in CIFAR-10.

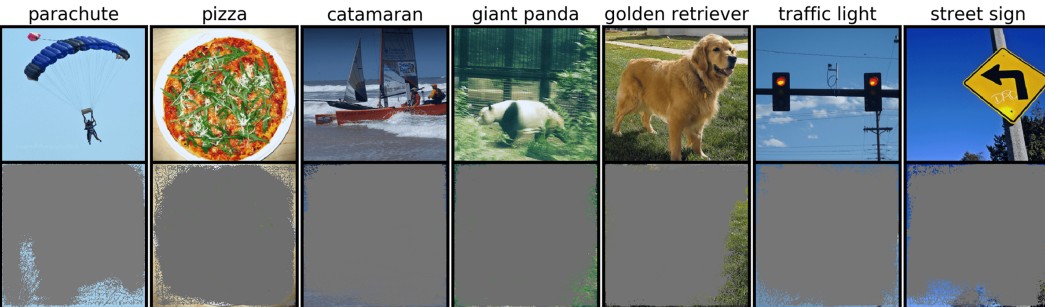

Figure 3: Sufficient input subsets (threshold 0.9) for example ImageNet validation images. The bottom row shows the corresponding images with all pixels outside of each SIS subset masked but are still classified by the Inception v3 model with $\geq 90\%$ confidence.

Table 1: Accuracy of CIFAR-10 classifiers trained and evaluated on full images, 5% backward selection (BS) pixel-subsets, and 5% random pixel-subsets. Where possible, we report accuracy as mean $\pm$ standard deviation (%) over five runs. For training on BS subsets, we run BS on all images for a single model of each type and average over five models trained on these subsets.

| Model | Train On | Evaluate On | CIFAR-10 Test Acc. | CIFAR-10-C Acc. |
|---|---|---|---|---|
| ResNet20 | Full Images | Full Images
5% BS Subsets
5% Random | $92.52 \pm 0.09$
$92.48$
$9.98 \pm 0.03$ | $69.44 \pm 0.52$
$70.65$
$10.02 \pm 0.01$ |
| | 5% BS Subsets | 5% BS Subsets | $92.49 \pm 0.02$ | $70.58 \pm 0.03$ |
| | 5% Random | 5% Random | $50.25 \pm 0.19$ | $44.04 \pm 0.33$ |
| | Input Dropout (Full) | Input Dropout (Full) | $91.02 \pm 0.25$ | $75.46 \pm 0.74$ |
| ResNet18 | Full Images | Full Images
5% BS Subsets
5% Random | $95.17 \pm 0.21$
$94.76$
$10.08 \pm 0.15$ | $75.08 \pm 0.20$
$75.15$
$10.08 \pm 0.07$ |
| | 5% BS Subsets | 5% BS Subsets | $94.96 \pm 0.04$ | $75.25 \pm 0.05$ |
| | 5% Random | 5% Random | $51.27 \pm 0.82$ | $45.24 \pm 0.45$ |
| | Input Dropout (Full) | Input Dropout (Full) | $94.15 \pm 0.26$ | $80.35 \pm 0.39$ |
| VGG16 | Full Images | Full Images
5% BS Subsets
5% Random | $93.69 \pm 0.12$
$93.27$
$10.02 \pm 0.18$ | $74.14 \pm 0.45$
$73.95$
$9.97 \pm 0.18$ |
| | 5% BS Subsets | 5% BS Subsets | $92.60 \pm 0.08$ | $73.27 \pm 0.18$ |
| | 5% Random | 5% Random | $53.66 \pm 1.96$ | $46.88 \pm 1.27$ |
| | Input Dropout (Full) | Input Dropout (Full) | $91.09 \pm 0.15$ | $80.43 \pm 0.24$ |
| Ensemble (ResNet18) | Full Images | Full Images
5% Random | $96.07$
$9.98$ | $77.00$
$10.01$ |

**ImageNet.** We also find models trained on ImageNet images suffer from severe overinterpretation. Figure 3 shows example SIS subsets (threshold 0.9) found via Batched Gradient SIS on images confidently classified by the pre-trained Inception v3 (additional examples in Figures S8 and S9). These SIS subsets appear visually nonsensical, yet the network classifies them with $\geq 90\%$ confidence. We find SIS pixels are concentrated outside of the actual object that determines the class label. For example, in the "pizza" image, the SIS is concentrated on the shape of the plate and the background table, rather than the pizza itself, suggesting the model could generalize poorly on images containing different circular items on a table. In the "giant panda" image, the SIS contains bamboo, which likely appeared in the collection of ImageNet photos for this class. In the "traffic light" and "street sign" images, the SIS consists of pixels in sky, suggesting that autonomous vehicle systems that may depend on these models should be carefully evaluated for overinterpretation pathologies.

Figure 2b shows SIS pixel locations from a random sample of 1000 ImageNet validation images. We find concentration along image borders, suggesting the model relies heavily on image backgrounds and suffers from severe overinterpretation. This is a serious problem as objects determining ImageNet classes are often located near image centers, and thus this network fails to focus on salient features.

## 4.2 Sparse Subsets are Real Statistical Patterns

The overconfidence of CNNs for image classification (Guo et al., 2017) may lead one to wonder whether the observed overconfidence on semantically meaningless SIS subsets is an artifact of calibration rather than true statistical signals in the dataset. We train models on 5% pixel-subsets of CIFAR-10 training images found via backward selection (Section 3.3). We find models trained solely on these pixel-subsets can classify corresponding test image pixel-subsets with minimal accuracy loss compared to models trained on full images (Table 1). As a baseline to the 5% pixel-subsets identified by backward selection, we create variants of all images where the 5% pixel-subsets are selected at

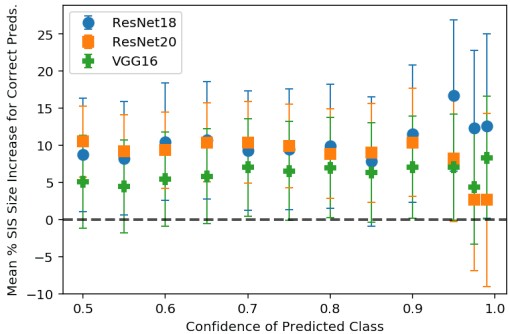 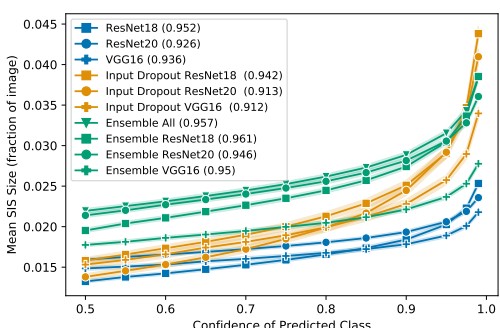

Figure 4: Percentage increase in mean SIS size of correctly classified compared to misclassified CIFAR-10 test images. Positive values indicate larger mean SIS size for correctly classified images. Error bars indicate 95% confidence interval for the difference in means.

Figure 5: Mean SIS size on CIFAR-10 test images as SIS threshold varies. SIS size indicates fraction of pixels necessary for model to make the same prediction at each confidence threshold. Model accuracies are shown in the legend. 95% confidence intervals are shaded around each mean.

random from each image (rather than by backward selection) and use the same random pixel-subsets for training each new model. Models trained on random subsets have significantly lower test accuracy (Table 1) compared to models trained on 5% pixel-subsets from backward selection. This result suggests that the highly sparse subsets found via backward selection offer a valid predictive signal in the CIFAR-10 benchmark exploited by models to attain high test accuracy. We observe similar results on ImageNet. Inception v3 trained on 10% pixel-subsets of ImageNet training images achieves 71.4% accuracy (mean over 5 runs) on the corresponding pixel-subset ImageNet validation set (Table S4).

### 4.3 HUMANS STRUGGLE TO CLASSIFY SPARSE SUBSETS

We find a strong correlation between the fraction of unmasked pixels in each image and human classification accuracy ($R^2 = 0.94$, Figure S7). Human accuracy on 5% pixel-subsets of CIFAR-10 images (mean = 19.2%, std dev = 4.8%, Table S3) is significantly lower than on original, unmasked images (roughly 94% (Karpathy, 2011)), though greater than random guessing, presumably due to correlations between labels and features such as color (e.g., blue sky suggests airplane, ship, or bird).

However, CNNs (even when trained on full images and achieve accuracy on par with human accuracy on full images) classify these sparse image subsets with very high accuracy (Table 1), indicating benchmark images contain statistical signals that are not salient to humans. Models solely trained to minimize prediction error may thus latch onto these signals while still accurately generalizing to test data, but may behave counterintuitively when fed images from a different source that does not share these exact statistics. The strong correlation between the size of CIFAR-10 pixel-subsets and the corresponding human classification accuracy suggests larger subsets contain more semantically salient content. Thus, a model whose decisions have larger corresponding SIS subsets presumably exhibits less overinterpretation than one with smaller SIS subsets, as we investigate in Section 4.4.

### 4.4 SIS SIZE IS PREDICTIVE OF MODEL ACCURACY

Given that smaller SIS contain fewer salient features according to human classifiers, models that justify their classifications based on sparse SIS subsets may be limited in terms of attainable accuracy, particularly in out-of-distribution settings. Here, we investigate the relationship between a model's predictive accuracy and the size of the SIS subsets in which it identifies class-evidence. For each of our three classifiers, we compute the average SIS size increase for correctly classified images as compared to incorrectly classified images (expressed as a percentage). We find SIS subsets of correctly classified images are consistently significantly larger than those of misclassified images at varying SIS confidence thresholds for both CIFAR-10 test images (Figure 4) and CIFAR-10-C OOD images (Figure S3). This is especially striking given model confidence is uniformly lower on the misclassified inputs (Figure S4). Lower confidence would normally imply a larger SIS subset at a

given confidence level, as one expects fewer pixels can be masked before the model's confidence drops below the SIS threshold. Thus, we can rule out overall model confidence as an explanation of the smaller SIS of misclassified images. This result suggests the sparse SIS subsets highlighted in this paper are not just a curiosity, but may be leading to poor generalization on real images.

## 4.5 MITIGATING OVERINTERPRETATION

**Ensembling.** Model ensembling is known to improve classification performance (Goh et al., 2001; Ju et al., 2018). As we found pixel-subset size to be strongly correlated with human pixel-subset classification accuracy (Section 4.3), our metric for measuring how much ensembling may alleviate overinterpretation is the increase in SIS subset size. We find ensembling uniformly increases test accuracy as expected but also increases the SIS size (Figure 5), hence mitigating overinterpretation.

We conjecture the cause of both the increase in the accuracy and SIS size for ensembles is the same. We observe that SIS subsets are generally not transferable from one model to another — i.e., an SIS for one model is rarely an SIS for another (Section 4.1). Thus, different models rely on different independent signals to arrive at the same prediction. An ensemble bases its prediction on multiple such signals, increasing predictive accuracy and SIS subset size by requiring simultaneous activation of multiple independently trained feature detectors. We find SIS subsets of the ensemble are larger than the SIS of its individual members (examples in Figure S2).

**Input Dropout.** We apply input dropout (Srivastava et al., 2014) to both train and test images. We retain each input pixel with probability $p = 0.8$ and set the values of dropped pixels to zero. We find a small decrease in CIFAR-10 test accuracy for models regularized with input dropout though find a significant ($\sim 6\%$) increase in OOD test accuracy on CIFAR-10-C images (Table 1, Figure S5). Figure 5 shows a corresponding increase in SIS subset size for these models, suggesting input dropout applied at train and test time helps to mitigate overinterpretation. We conjecture that random dropout of input pixels disrupts spurious signals that lead to overinterpretation.

## 5 DISCUSSION

We find that modern image classifiers overinterpret small nonsensical patterns present in popular benchmark datasets, identifying strong class evidence in the pixel-subsets that constitute these patterns. Despite their lack of salient features, these sparse pixel-subsets are underlying statistical signals that suffice to accurately generalize from the benchmark training data to the benchmark test data. We found that different models rationalize their predictions based on different sufficient input subsets, suggesting optimal image classification rules remain highly underdetermined by the training data. In high-stakes applications, we recommend ensembles of networks or regularization via input dropout.

Our results call into question model interpretability methods whose outputs are encouraged to align with prior human beliefs regarding proper classifier operating behavior (Adebayo et al., 2018). Given the existence of non-salient pixel-subsets that alone suffice for correct classification, a model may solely rely on these patterns. In this case, an interpretability method that faithfully describes the model should output these nonsensical rationales, whereas interpretability methods that bias rationales toward human priors may produce results that mislead users to think their models behave as intended.

Mitigating overinterpretation and the broader task of ensuring classifiers are accurate for the right reasons remain significant challenges for ML. While we identify strategies for partially mitigating overinterpretation, additional research needs to develop ML methods that rely exclusively on well-formed interpretable inputs, and methods for creating training data that do not contain spurious signals. One alternative is to regularize CNNs by constraining the pixel attributions generated via a saliency map (Ross et al., 2017; Simpson et al., 2019; Viviano et al., 2019). Unfortunately, such methods require a human annotator to highlight the correct pixels as an auxiliary supervision signal. Saliency maps have also been shown to provide unreliable insights into model operating behavior and must be interpreted as approximations (Kindermans et al., 2019). In contrast, our SIS subsets constitute actual pathological examples that have been misconstrued by the model. An important application of our methods is the evaluation of training datasets to ensure decisions are made on interpretable rather than spurious signals. We found popular image datasets contain such spurious signals, and the resulting overinterpretation may be difficult to overcome with ML methods alone.

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

# Supplementary Information:
# Overinterpretation reveals image classification model pathologies

## S1 MODEL IMPLEMENTATION AND TRAINING DETAILS

### CIFAR-10 MODELS

We first describe the implementation and training details for the CIFAR-10 models used in this paper (Section 3.1). The ResNet20 architecture (He et al., 2016a) has 16 initial filters and a total of 0.27M parameters. ResNet18 (He et al., 2016b) has 64 initial filters and contains 11.2M parameters. The VGG16 architecture (Simonyan & Zisserman, 2014) uses batch normalization and contains 14.7M parameters.

All models are trained for 200 epochs with a batch size of 128. We minimize cross-entropy via SGD with Nesterov momentum (Sutskever et al., 2013) using momentum of 0.9 and weight decay of 5e-4. The learning rate is initialized as 0.1 and is reduced by a factor of 5 after epochs 60, 120, and 160. Datasets are normalized using per-channel mean and standard deviation, and we use standard data augmentation strategies consisting of random crops and horizontal flips (He et al., 2016b).

The adversarially robust model we evaluated is the `adv_trained` model of Madry et al. (2017), available on GitHub[1].

To apply the SIS procedure to CIFAR-10 images, we use an implementation available on GitHub[2]. For confidently classified images on which we run SIS, we find one sufficient input subset per image using the FindSIS procedure. When masking pixels, we mask all channels of each pixel as a single feature.

### IMAGENET MODELS

For finding SIS, we use pre-trained models (Inception v3 (Szegedy et al., 2016) and ResNet50 (He et al., 2016a)) provided by PyTorch (Paszke et al., 2019) in the torchvision package (PyTorch version 1.4.0, torchvision version 0.5.0).

When training new ImageNet classifiers, we adopt model implementations and training scripts from PyTorch (Paszke et al., 2019), obtained from GitHub[3]. Models are trained for 90 epochs using batch size 256 (Inception-v3) or 512 (ResNet50). We minimize cross-entropy via SGD using momentum of 0.9 and weight decay of 1e-4. The learning rate is initialized as 0.1 and reduced by a factor of 10 every 30 epochs. Datasets are normalized using per-channel mean and standard deviation. For Inception v3, images are cropped to 299 x 299 pixels. For ResNet50, images are cropped to 224 x 224. When training Inception v3, we define the model using the `aux_logits=False` argument. We do not use data augmentation when training models on pixel-subsets of images.

### HARDWARE DETAILS

Each CIFAR-10 model is trained on 1 NVIDIA GeForce RTX 2080 Ti GPU. Once models are trained, SIS are computed across multiple GPUs (by parallelizing over individual images). Each SIS (for 1 CIFAR-10 image) takes roughly 30-60 seconds to compute (depending on the model architecture).

ImageNet models are trained on 2 NVIDIA Titan RTX GPUs. For finding SIS from pre-trained ImageNet models, we run Batched Gradient BackSelect for batches of 32 images across 10 NVIDIA GeForce RTX 2080 Ti GPUs, which takes roughly 1-2 minutes per batch (details in Section S5).

---

[1] `https://github.com/MadryLab/cifar10_challenge`
[2] `https://github.com/google-research/google-research/blob/master/sufficient_input_subsets/sis.py`
[3] `https://github.com/pytorch/examples/blob/master/imagenet/main.py`

## S2 Additional Examples of CIFAR-10 Sufficient Input Subsets

### SIS of Individual Networks

Figure S1 shows a sample of SIS for each of our three architectures. These images were randomly sampled among all CIFAR-10 test images confidently (confidence $\geq 0.99$) predicted to belong to the class written on the left. SIS are computed under a threshold of 0.99, so all images shown in this figure are classified with probability $\geq 99\%$ confidence as belonging to the listed class.

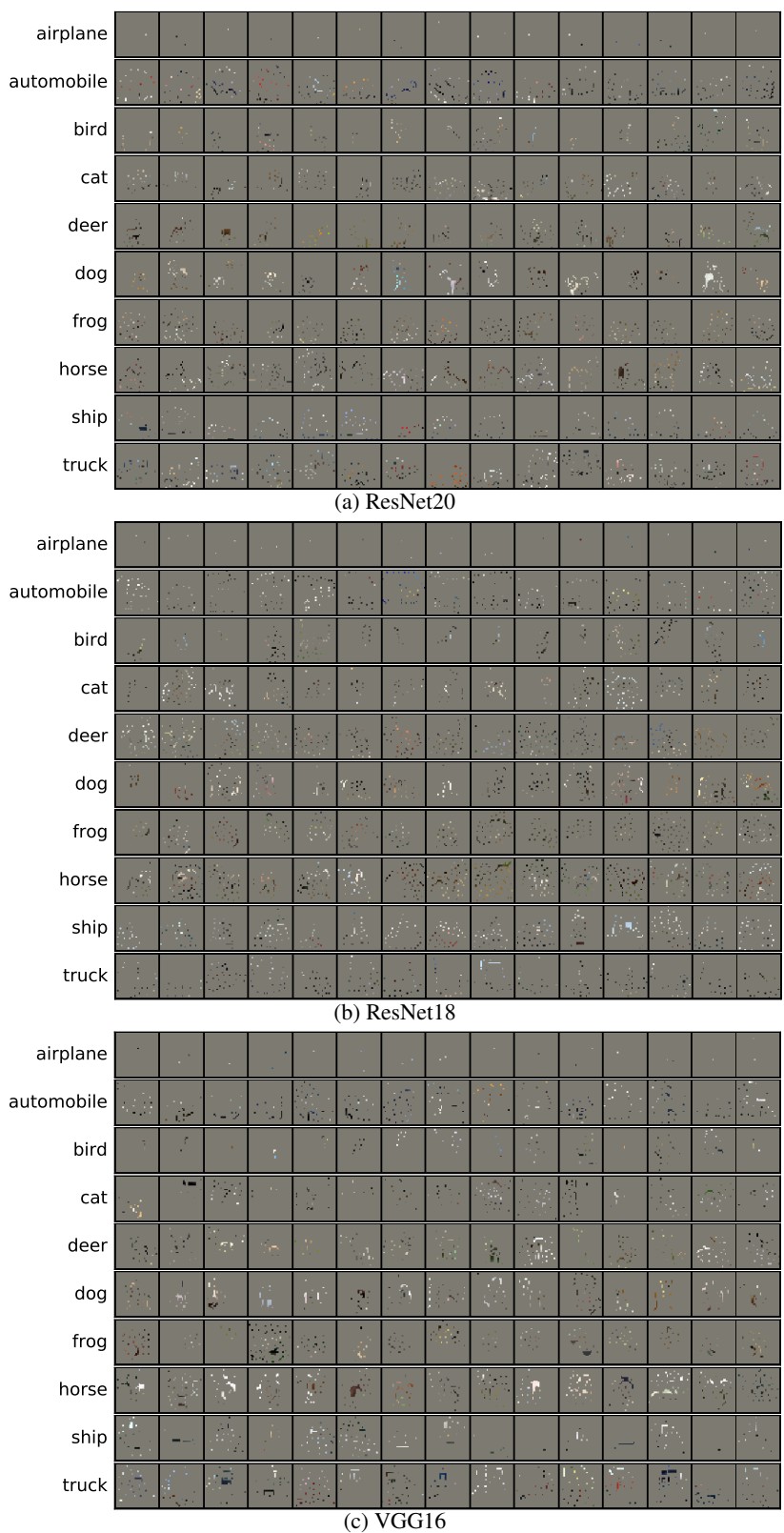

(a) ResNet20

(b) ResNet18

(c) VGG16

Figure S1: Examples of SIS (threshold 0.99) on random sample of CIFAR-10 test images (15 per class, different random sample for each architecture). All images shown here are predicted to belong to the listed class with $\geq 99\%$ confidence.

SIS OF ENSEMBLE

Figure S2 shows examples of SIS from one of our model ensembles (a homogeneous ensemble of ResNet18 networks, see Section 3.1), along with corresponding SIS for the same image from each of the five member networks in the ensemble. We use a SIS threshold of 0.99, so all images are classified with $\geq 99\%$ confidence. These examples highlight how the ensemble SIS are larger and draw class-evidence from the individual members' SIS.

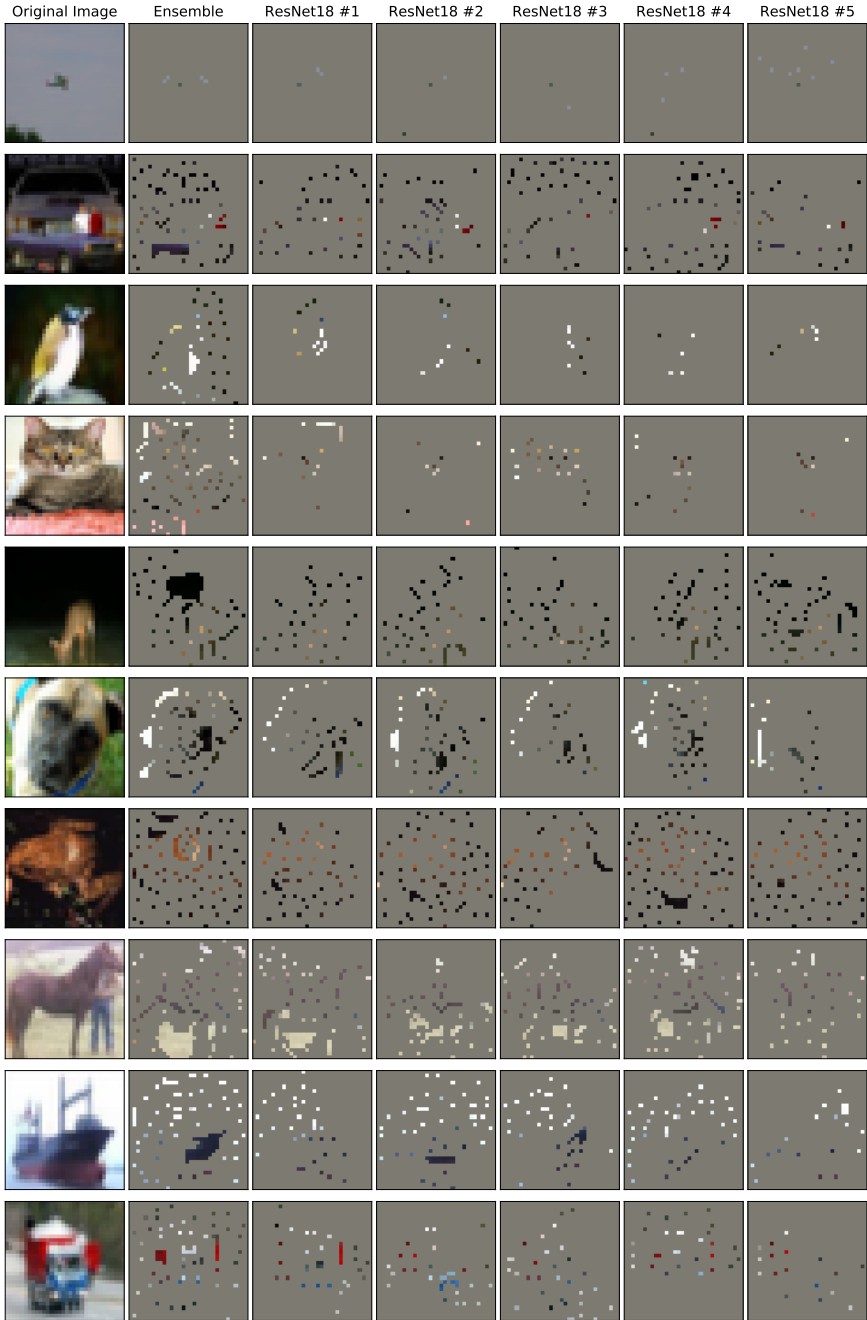

Figure S2: Examples of SIS (threshold 0.99) from the ResNet18 homogeneous ensemble (Section 3.1) and its member models. Each row shows original CIFAR-10 image (left), followed by SIS from the ensemble (second column) and the SIS from each of its 5 member networks (remaining columns). Each image shown is classified with $\geq 99\%$ confidence by its respective network.

## S3   ADDITIONAL MODEL PERFORMANCE RESULTS

### TRAINING ON PIXEL-SUBSETS WITH DATA AUGMENTATION

Table S1 presents results similar to those in Section 4.2 and Table 1, but where models are trained on 5% pixel-subsets with data augmentation (as described in Section S1). We find training without data augmentation slightly improves accuracy when training classifiers on 5% pixel-subsets of CIFAR-10.

Table S1: Accuracy of CIFAR-10 classifiers trained and evaluated on full images, 5% backward selection (BS) pixel-subsets, and 5% random pixel-subsets *with* data augmentation (+). Accuracy is reported as mean $\pm$ standard deviation (%) over five runs.

| Model | Train On | Evaluate On | CIFAR-10 Test Acc. | CIFAR-10-C Acc. |
|---|---|---|---|---|
| ResNet20 | 5% BS Subsets (+) | 5% BS Subsets | $92.26 \pm 0.01$ | $70.21 \pm 0.14$ |
|  | 5% Random (+) | 5% Random | $48.87 \pm 0.41$ | $42.66 \pm 0.15$ |
| ResNet18 | 5% BS Subsets (+) | 5% BS Subsets | $94.51 \pm 0.38$ | $74.91 \pm 0.41$ |
|  | 5% Random (+) | 5% Random | $49.03 \pm 0.92$ | $42.97 \pm 0.82$ |
| VGG16 | 5% BS Subsets (+) | 5% BS Subsets | $91.17 \pm 0.04$ | $71.82 \pm 0.13$ |
|  | 5% Random (+) | 5% Random | $51.32 \pm 1.35$ | $44.56 \pm 0.96$ |

### TRAINING ON PIXEL-SUBSETS WITH DIFFERENT ARCHITECTURES

Table S2 presents results of training and evaluating models on 5% pixel-subsets drawn from different architectures. Models were trained without data augmentation on subsets from one replicate of each base architecture. We find accuracy from training and evaluating a model on 5% pixel-subsets of images derived from a different architecture is commensurate with accuracy of training and evaluating a new model of the same type on those subsets (Table 1).

Table S2: Accuracy of CIFAR-10 classifiers trained and evaluated on 5% backward selection (BS) pixel-subsets from different architectures. Accuracy is reported as mean $\pm$ standard deviation (%) over five runs.

| 5% Subsets from Model | Model Trained | CIFAR-10 Test Acc. | CIFAR-10-C Acc. |
|---|---|---|---|
| ResNet20 | ResNet18 | $92.53 \pm 0.02$ | $70.56 \pm 0.04$ |
|  | VGG16 | $92.47 \pm 0.02$ | $70.42 \pm 0.14$ |
| ResNet18 | ResNet20 | $94.88 \pm 0.03$ | $75.14 \pm 0.10$ |
|  | VGG16 | $94.88 \pm 0.05$ | $75.13 \pm 0.09$ |
| VGG16 | ResNet20 | $92.05 \pm 0.14$ | $73.01 \pm 0.08$ |
|  | ResNet18 | $92.57 \pm 0.10$ | $73.33 \pm 0.21$ |

### ADDITIONAL RESULTS FOR SIS SIZE AND MODEL ACCURACY

Figure S3 shows percentage increase in mean SIS size for correctly classified images compared to misclassified images from the CIFAR-10-C dataset.

Figure S4 shows the mean confidence of each group of correctly and incorrectly classified images that we consider at each confidence threshold (at each confidence threshold along the x-axis, we evaluate SIS size in Figure 4 on the set of images that originally were classified with at least that level of confidence). We find model confidence is uniformly lower on the misclassified inputs.

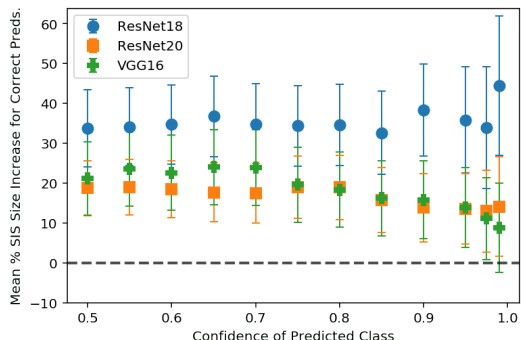

Figure S3: Percentage increase in mean SIS size of correctly classified images compared to misclassified images from a random sample of CIFAR-10-C test set. Positive values indicate larger mean SIS size for correctly classified images. Error bars indicate 95% confidence interval for the difference in means.

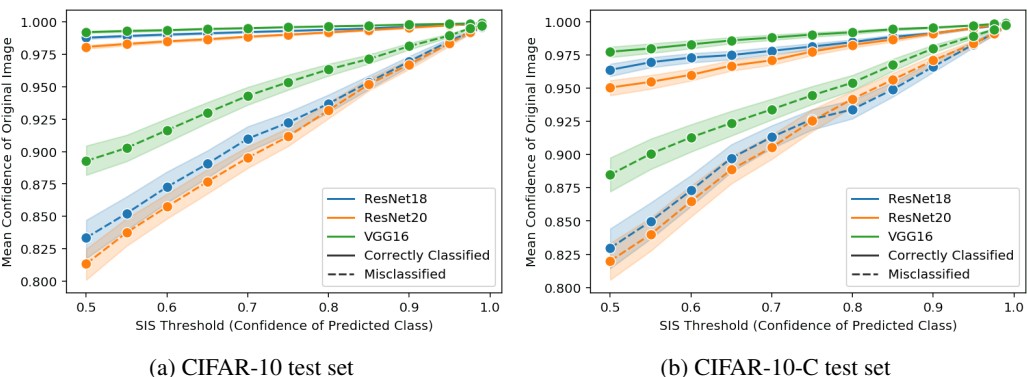

(a) CIFAR-10 test set        (b) CIFAR-10-C test set

Figure S4: Mean confidence of correctly vs. incorrectly classified images for each corresponding SIS threshold we evaluate in Figure 4 across the (a) CIFAR-10 test set and (b) our random sample of the CIFAR-10-C test set. Shaded region indicates 95% confidence interval.

ADDITIONAL RESULTS FOR INPUT DROPOUT

Figure S5 shows the accuracy improvement on each individual corruption of the CIFAR-10-C out-of-distribution test set for models trained with input dropout (Section 4.5) compared to original models.

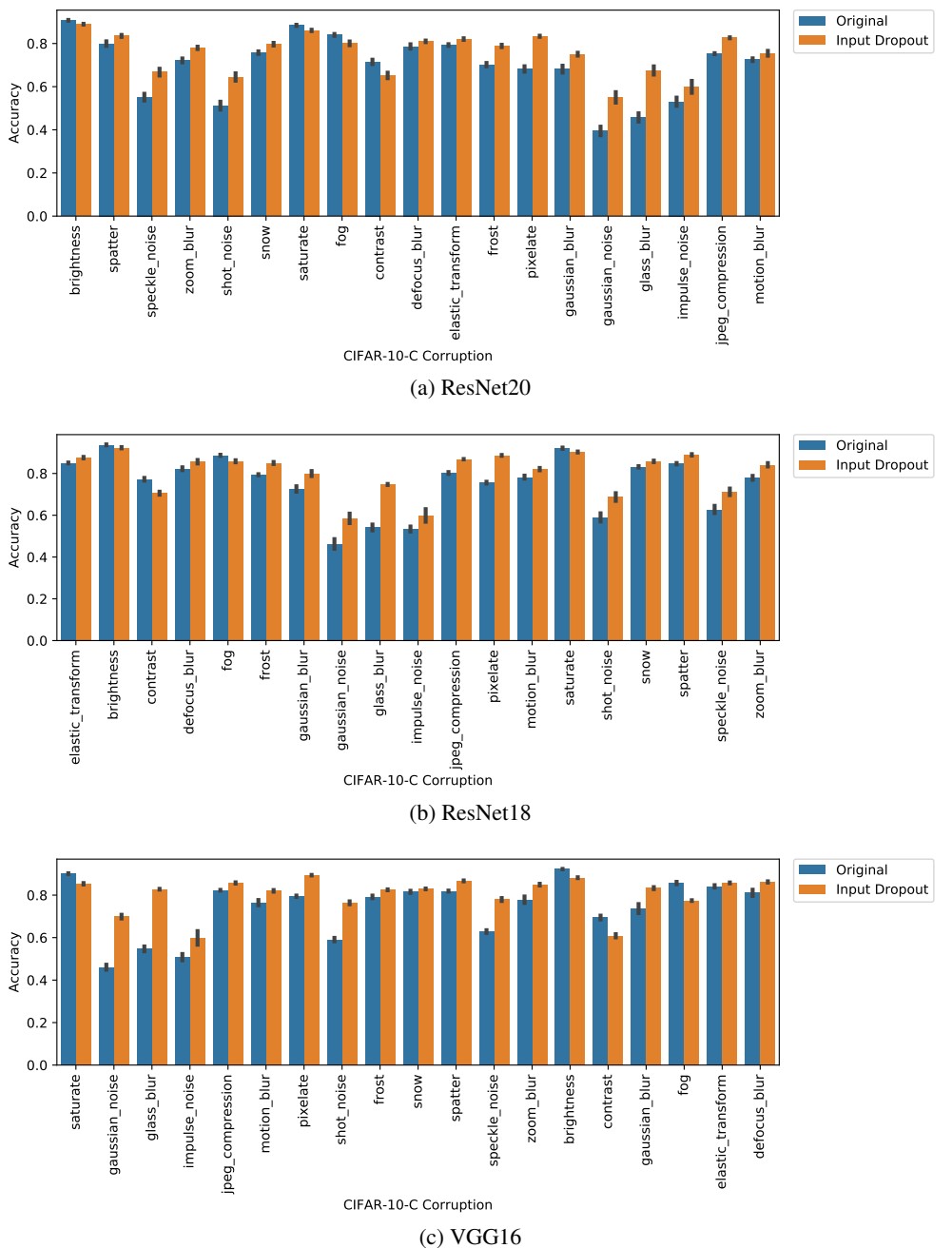

(a) ResNet20

(b) ResNet18

(c) VGG16

Figure S5: Accuracy on individual corruptions of CIFAR-10-C out-of-distribution images for original models and models trained with input dropout (Section 4.5). Accuracy is given as mean ± standard deviation over five replicate models.

## S4 Details of Human Classification Benchmark

Here we include additional details on our benchmark of human classification accuracy of sparse pixel-subsets (Section 3.4). Figure S6 shows all images shown to users (100 images each for 5%, 30% and 50% pixel-subsets of CIFAR-10 test images). Each set of 100 images has pixel-subsets stemming from each of the three architectures roughly equally (35 ResNet20, 35 ResNet18, 30 VGG16).[4] Figure S7 shows the correlation between human classification accuracy and pixel-subset size (accuracies shown in Table S3).

Table S3: Human classification accuracy on a sample of CIFAR-10 test image pixel-subsets of varying sparsity (see Section 3.4). Accuracies given as mean $\pm$ standard deviation.

| Fraction of Images | Human Classification Accuracy (%) |
|---|---|
| 5% | $19.2 \pm 4.8$ |
| 30% | $40.0 \pm 2.5$ |
| 50% | $68.2 \pm 3.6$ |

---

[4]The human classification benchmark was performed using pixel-subsets computed from earlier implementations of the three CNN architectures (in Keras rather than PyTorch). Figure S5 shows all pixel-subsets derived from these models that were shown to users in the human classification benchmark. ResNet20 was based on a Keras example using 16 initial filters and optimized with Adam for 200 epochs (batch size 32, initial learning rate 0.001, reduced after epochs 80, 120, 160, and 180 to 1e-4, 1e-5, 1e-6, and 5e-7, respectively). ResNet18 was based on a GitHub implementation using 64 initial filters, initial strides (1, 1), initial kernel size (3, 3), no initial pooling layer, weight decay 0.0005 and trained using SGD with Nesterov momentum 0.9 for 200 epochs (batch size 128, initial learning rate 0.1, reduced by a factor of 5 after epochs 60, 120, and 160). VGG16 was based on a GitHub implementation trained with weight decay 0.0005 and SGD with Nesterov momentum 0.9 for 250 epochs (batch size 128, initial learning rate 0.1, decayed after each epoch as $0.1 \cdot 0.5^{\lfloor \text{epoch}/20 \rfloor}$). We selected the final model checkpoint that maximized test accuracy. We found these models exhibited similar overinterpretation behavior to the final models.

- https://keras.io/examples/cifar10_resnet/
- https://github.com/keras-team/keras-contrib/blob/master/keras_contrib/applications/resnet.py
- https://github.com/geifmany/cifar-vgg/blob/e7d4bd4807d15631177a2fafabb5497d0e4be3ba/cifar10vgg.py

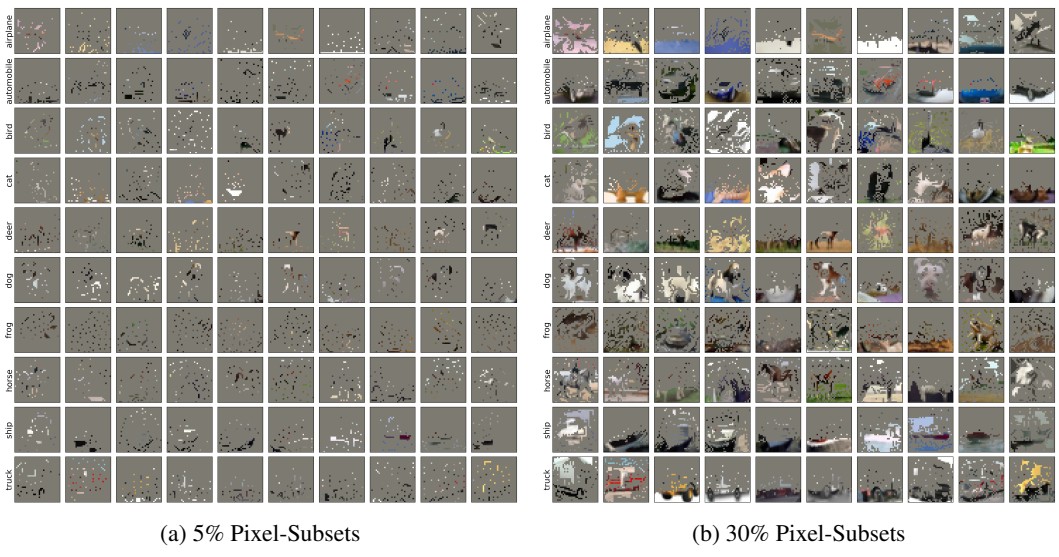

(a) 5% Pixel-Subsets

(b) 30% Pixel-Subsets

(c) 50% Pixel-Subsets

Figure S6: Pixel-subsets of CIFAR-10 test images shown to participants in our human classification benchmark (Section 3.4).

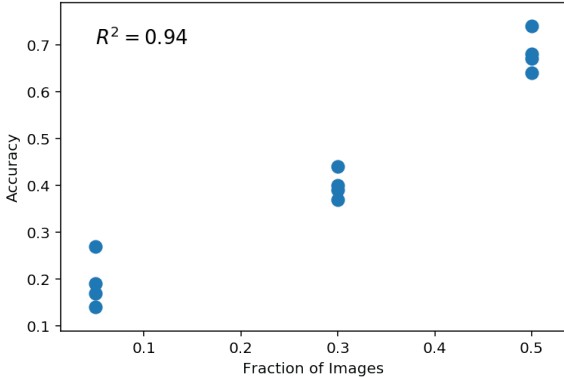

Figure S7: Human classification accuracy on a sample of CIFAR-10 test image pixel-subsets (see Section 3.4).

## S5  DETAILS OF BATCHED GRADIENT SIS ALGORITHM

It is computationally infeasible to scale the original backward selection procedure of SIS (Carter et al., 2019) to ImageNet. As each ImageNet image contains $299 \times 299 = 89401$ pixels, running backward selection to find one SIS for an image would require $\sim 4$ billion forward passes through the network. Here we introduce a more efficient gradient-based approximation to the original SIS procedure (via **Batched Gradient SIScollection**, **Batched Gradient BackSelect**, and **Batched Gradient FindSIS**) that allows us to find SIS on larger ImageNet images in a reasonable time. The **Batched Gradient SIScollection** procedure described below identifies a complete collection of disjoint masks for an input $\mathbf{x}$, where each mask $M$ specifies a pixel-subset of the input $\mathbf{x}_S = \mathbf{x} \odot (1 - M)$ such that $f(\mathbf{x}_S \geq \tau)$. Here $f$ outputs the probability assigned by the network to its predicted class (i.e., its confidence).

The idea behind our approximation algorithm is two-fold: (1) Instead of separately masking every remaining pixel to find the least critical pixel (whose masking least reduces the confidence in the network's prediction), we use the *gradient* with respect to the mask as a means of ordering. (2) Instead of masking just 1 pixel per iteration, we mask larger subsets of $k \geq 1$ pixels per iteration. More formally, let $\mathbf{x}$ be an image of dimensions $H \times W \times C$ where $H$ is the height, $W$ the width, and $C$ the channel. Let $f(\mathbf{x})$ be the network's confidence on image $\mathbf{x}$ and $\tau$ the target SIS confidence threshold. Recall that we only compute SIS for images where $f(\mathbf{x}) \geq \tau$. Let $M$ be the mask with dimensions $H \times W$ with 0 indicating an unmasked feature (pixel) and 1 indicating a masked feature. We initialize $M$ as all 0s (all features unmasked). At iteration $i$, we compute the gradient of $f$ with respect to the input pixels and mask $\nabla M = \nabla_M f(\mathbf{x} \odot (1 - M))$. Here $M$ is the current mask updated after each iteration. In each iteration, we find the block of $k$ features to mask, $G^*$, chosen in descending order by value of entries in $\nabla M$. The mask is updated after each iteration by masking this block of $k$ features until all features have been masked. Given $p$ input features, our **Batched Gradient SIScollection** procedure returns $j$ sufficient input subsets in $\mathcal{O}(\frac{p}{k} \cdot j)$ evaluations of $\nabla f$ (as opposed to $\mathcal{O}(p^2 j)$ evaluations of $f$ in the original SIS procedure (Carter et al., 2019)).

We use $k = 100$ in this paper, which allows us to find one SIS for each of 32 ImageNet images (i.e., a mini-batch) in $\sim$1-2 minutes using **Batched Gradient FindSIS**. Note that while our algorithm is an approximate procedure, the pixel-subsets produced are real sufficient input subsets, that is they always satisfy $f(\mathbf{x}_S \geq \tau)$. For CIFAR-10 images (which are smaller in size), we use the original SIS procedure from (Carter et al., 2019). For both datasets, we treat all channels of each pixel as a single feature.

---

**Batched Gradient SIScollection**($f, \mathbf{x}, \tau, k$)

---

$M = \mathbf{0}$
**for** $j = 1, 2, \ldots$ **do**
    | $R = $ **Batched Gradient BackSelect**($f, \mathbf{x}, M, k$)
    | $M_j = $ **Batched Gradient FindSIS**($f, \mathbf{x}, \tau, R$)
    | $M \leftarrow M + M_j$
    | **if** $f(\mathbf{x} \odot (1 - M)) < \tau$: **return** $M_1, \ldots, M_{j-1}$
**end**

---

**Batched Gradient BackSelect**($f, \mathbf{x}, M, k$)

---

$R = $ empty stack
**while** $M \neq \mathbf{1}$ **do**
    | $G^* = \mathrm{Top}_k (\nabla_M f(\mathbf{x} \odot (1 - M))$
    | Update $M \leftarrow M + G^*$
    | Push $G^*$ onto top of $R$
**end**
**return** $R$

---

---

**Batched Gradient FindSIS**($f$, **x**, $\tau$, $R$)

---

$M = \mathbf{1}$
**while** $f(\mathbf{x} \odot (1 - M)) < \tau$ **do**
   | Pop $G$ from top of $R$
   | Update $M \leftarrow M - G$
**end**
**if** $f(\mathbf{x} \odot (1 - M)) \geq \tau$: **return** $M$
**else**: **return** *None*

---

## S6    Additional Results of ImageNet Overinterpretation

### Training CNNs on ImageNet Pixel-Subsets

We extracted 10% pixel-subsets by applying Batched Gradient BackSelect to all ImageNet train and validation images using pre-trained Inception v3 and ResNet50 models from PyTorch (Paszke et al., 2019). We kept the top 10% of pixels and masked the remaining 90% with zeros. We trained new models of the same type on these 10% pixel-subsets of ImageNet training set images (training details in Section S1) and evaluated the resulting models on the corresponding 10% pixel-subsets of ImageNet validation images. Table S4 shows a small loss in validation accuracy, suggesting these 10% pixel-subsets that are indiscernible by humans contain statistically valid signals that generalize to validation images. Models trained on 10% pixel-subsets were trained without data augmentation. As with CIFAR-10 (Section S3), we found training models on pixel-subsets with standard data augmentation techniques (random crops and horizontal flips) resulted in worse validation accuracy.

Table S4: Accuracy of models on ImageNet validation images trained and evaluated on full images and 10% BS pixel-subsets. Accuracy for training on 10% BS Subsets is reported as mean $\pm$ standard deviation (%) over five training runs with different random initialization. For training/evaluation on BS pixel-subsets, we run backward selection on all ImageNet images using a single pre-trained model of each type, but average over five models trained on these subsets.

| Model | Train On | Evaluate On | Top 1 Acc. | Top 5 Acc. |
|-------|----------|-------------|------------|------------|
| Inception v3 | Full Images (pre-trained) | Full Images | 77.21 | 93.53 |
| | | 10% BS Subsets | 73.87 | 83.43 |
| | 10% BS Subsets | 10% BS Subsets | $71.37 \pm 0.15$ | $83.73 \pm 0.10$ |
| ResNet50 | Full Images (pre-trained) | Full Images | 76.13 | 92.86 |
| | | 10% BS Subsets | 45.14 | 64.12 |
| | 10% BS Subsets | 10% BS Subsets | $65.71 \pm 0.08$ | $80.45 \pm 0.08$ |

### Additional Examples of SIS on ImageNet

Figure S8 shows additional examples of SIS (threshold 0.9) on ImageNet validation images from the pre-trained Inception v3 found via Batched Gradient FindSIS.

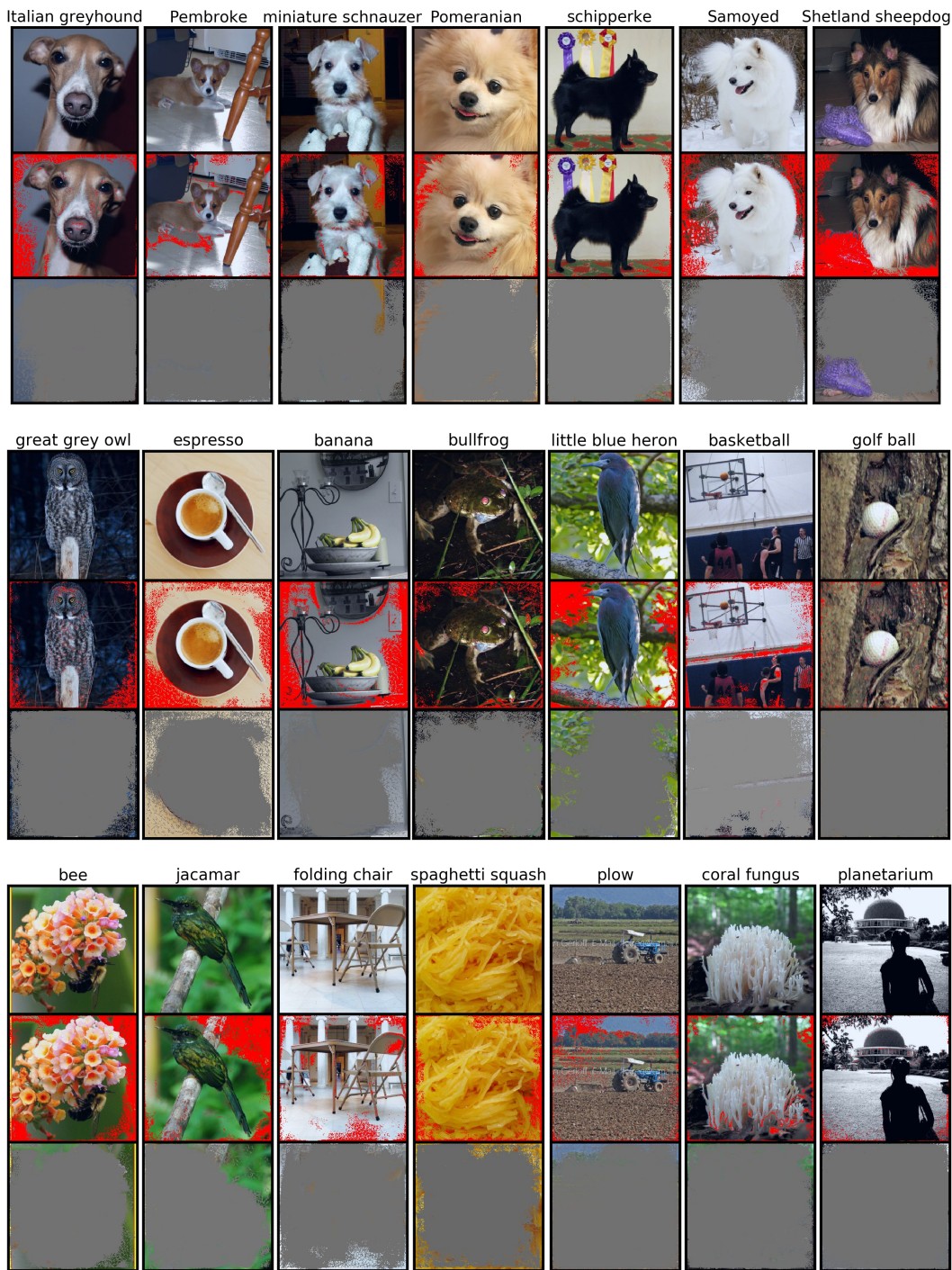

Figure S8: Example SIS (threshold 0.9) from ImageNet validation images (top row of each block). The middle rows show the location of SIS pixels (red) and the bottom rows show images with all non-SIS pixels masked but are still classified by the Inception v3 model with $\geq 90\%$ confidence.

We also explored the relationship between pixel saliency and the order pixels were removed by Batched Gradient BackSelect. Surprisingly, as shown in Figure S9 for Inception v3, we found that the most salient pixels were often *eliminated first* and thus unnecessary for maintaining high predicted confidence on the remaining pixel-subsets and subsequently for training on pixel-subsets. Figure S10 shows the predicted confidence on remaining pixels at each step of the Batched Gradient BackSelect procedure for a random sample of 32 ImageNet validation images by the Inception v3 model.

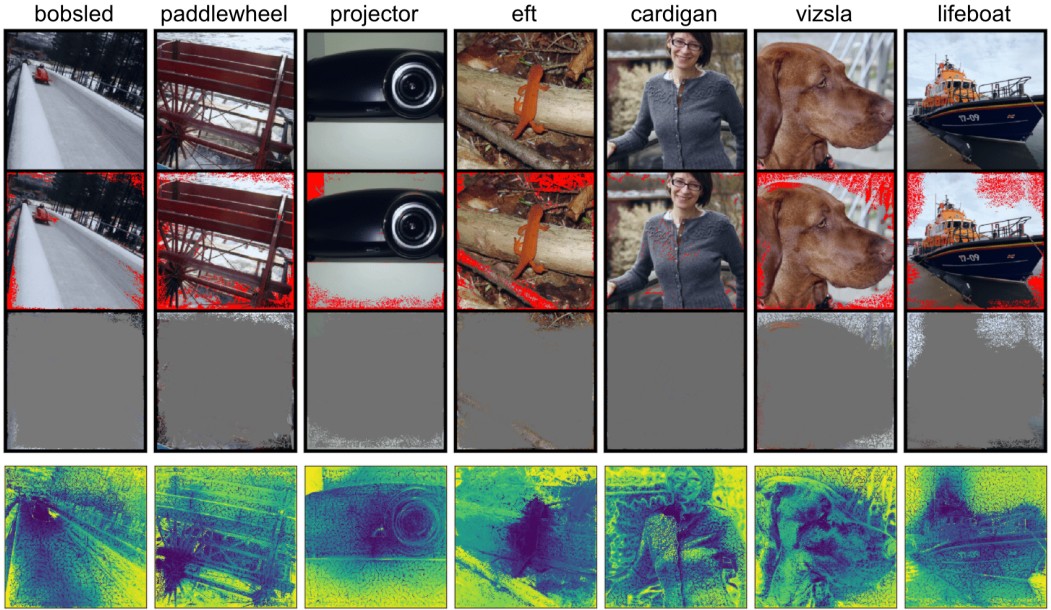

Figure S9: SIS subsets and ordering of pixels removed by Batched Gradient FindSIS in a sample of ImageNet validation images that are confidently ($\geq 90\%$) and correctly classified by the Inception v3 model. The top row shows original images, second row shows the location of SIS pixels (red), and third row shows images with all non-SIS pixels masked (and are still classified correctly with $\geq 90\%$ confidence). The heatmaps in the bottom row depict the ordering of batches of pixels removed during backward selection (blue = earliest, yellow = latest).

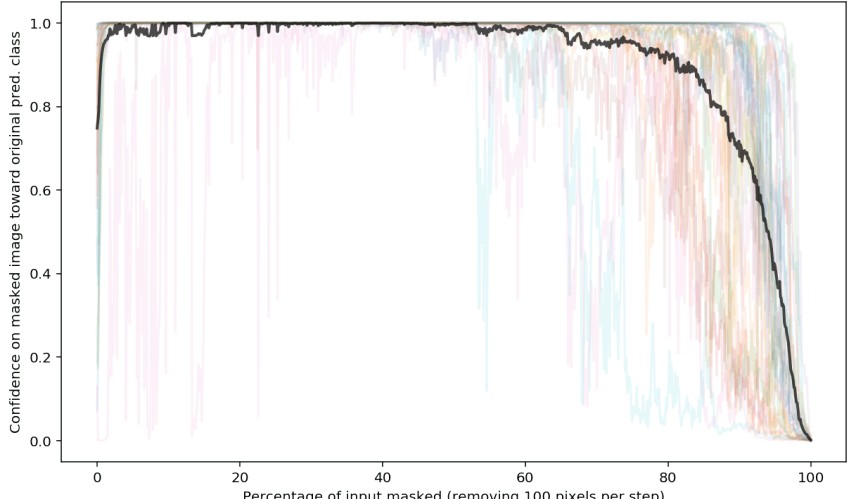

Figure S10: Prediction history on remaining (unmasked) pixels at each step of the Batched Gradient BackSelect procedure for a random sample of 32 ImageNet validation images by the Inception v3 model. Black line depicts mean confidence at each step.

