# OpenReview forum: "Overinterpretation reveals image classification model pathologies"
_ICLR.cc/2021/Conference — Reject_

### Official Review · AnonReviewer4 · 2020-10-25
**Interesting empirical results on “overinterpretation” of DNNs with a caveat**

**Rating:** 5
**Confidence:** 4

**Review:**

Summary:
This paper studies “overinterpretation”(provides a high-confidence decision without
salient supporting input features) of deep learning models. It modifies previous method Sufficient Input Subsets (SIS) (Carter et al., 2019) to scale to high-dimensional inputs. It detects overinterpretation on both CIFAR10 and ImageNet. In particular, when masking 90% of original pixels, the model can still achieve high accuracy. It then proposes two strategies (ensembles and dropout) to mitigate such overinterpretation artifacts.

################################################

Reasons for score:
The paper is overall well-written and presents some interesting findings on the so-called “overinterpretation” of DNNs classifiers. The scaled version of SIS and the usage of ensembles / dropout to mitigate overinterpretation is intuitive although not novel. However, I have a concern regarding its experimental results and consequences.

################################################

Pros:

+mostly well-written

+important topic and interesting findings

+extensive experiments

+code is provided

Cons:

-No results are shown on training on sparse pixels and evaluating on full images

This is my biggest concern. This experiment seems important since otherwise it is not convincing that the model only uses those remaining pixels to make the decision. I am thus not fully convinced by the statement “We show misclassifications often rely on smaller and more spurious feature subsets suggesting overinterpretation is a serious practical issue”.

-The consequence of the findings needs a bit more discussion. How can one benefit from reducing the “overinterpretation” in practice?

-Only empirical results are presented and not enough discussions on the theoretical side.

-Another minor concern is about the usage of mean SIS size for measuring semantic meaning. It seems to be a reasonable proxy. However, I think it might be more useful to look at the mean SIS size of those that are wrongly classified by humans. If an image is already correctly classified by humans (there are indeed some figures that show the semantic features of the original objects despite only 5-10% pixels showing up, see e.g. Figure 1 bottom row the third to last image shows the contour of a horse), the increase of SIS size may not add much more semantic meaning.

################################################

Minor:

-last paragraph 1st page “although they propose differing explanations for the decisions of a model”, “differing” should be “different”

-sec 3.2 ImageNet, the mask “M” is not defined

################################################

Questions:

-I am really surprised by the performance of training on 5% random and evaluating on 5% random (around 50% as shown in table 1). What can be a possible explanation here?

-cifar10-c adds visual effect to images. It might be more interesting to also show the performance on spatially transformed images (those should be heavily influenced when only sparse original pixels are used).



################################################

Post-Rebuttal:

Thanks the authors for their detailed response! After reading the responses, I decide to maintain my initial assessment.

The statement "full images are highly out-of-distribution for a model trained on images with only 5% unmasked pixel-subsets and hence such a model cannot properly generalize to fully unmasked images" makes sense. However, I still feel that the authors need an experiment of this flavor to support their claim of “We show misclassifications often rely on smaller and more spurious feature subsets suggesting overinterpretation is a serious practical issue” as mentioned in my initial review as well as pointed out by R1.

Besides, I also find the point "the observed phenomenon is very model-dependent" raised by R1 is a valid major concern. In the authors response, they did not add extra experiments to address it. " We indeed find that models trained on 5% pixel-subsets can generalize to the corresponding 5% pixel-subsets of test images. " - the stated experiment trains and tests on the same model so it does not address the concern that the observed phenomenon is model-dependent. In order to address this concern, the authors need to add some experiments on transferring across architectures (e.g. train on SIS of ResNet and test on SIS of VGG).

---

> ### Author Response · Authors · 2020-11-23
> **Response to Reviewer 4**
>
> We thank the reviewer for the comments and encouraging feedback.
>
> **[Train models on 5% subsets and test on full images]**  Please see our response to R1 regarding this experiment above.
>
> **[Performance of models trained on 5% random subsets]**  We agree that this is an interesting finding, and further reinforces the caution that should be used in interpreting results from these popular benchmark datasets.  It is another indication of the degeneracy of these datasets and is a troubling finding of the paper that random selections of downsampled pixels actually enable classification 5x better than random guessing.
>
> **[Performance on spatially transformed images]**   Thanks for this suggestion.  The focus of this paper is to alert the community to issues with untransformed popular benchmark datasets.  We wanted to show the pathologies that result from unmodified subsets of original images and the ability of these signals to generalize to the held-out sets of conventional benchmarks.

---

### Official Review · AnonReviewer3 · 2020-10-28
**A below-borderline work that lacks novelty.**

**Rating:** 2
**Confidence:** 4

**Review:**

##########################################################################

Summary:

This paper proposes "overinterpretation" which describes the phenomenon that CNNs could achieve high test accuracy while replying on features that lack semantic meaning. To demonstrate overinterpretation on CIFAR-10 and ImageNet, the authors use Batched Gradient SIS to select a small subset of pixels for each image and trained CNNs on the modified images. While humans can not make accurate predictions on those modified images, CNNs can still achieve high test accuracy. Lastly, the authors propose to use ensembling and input dropout to address overinterpretation.

##########################################################################

Reasons for score:

Overall, I vote for rejecting. The main reason is that this paper lacks novelty. While the authors pointed out a realistic issue of CNNs that they could rely on irrelevant features for predictions, this drawback has been largely explored in previous works. Furthermore, the solutions (simple ensembling & input dropout ) proposed by the authors have also been thoroughly studied in previous works.

##########################################################################

Pros:

This paper points out a realistic pathology that is shared by mainstream CNN architectures.

The paper is well-organized.


##########################################################################

Cons:

This paper lacks novelty. Both the phenomenon of "overinterpretation" and the proposed solutions have been thoroughly studied in previous works.

While the authors show that CNNs could rely on irrelevant features, they did not investigate the cause of their behavior. Specifically, the paper does not investigate whether such pathological behavior is caused by the properties of the datasets or it's originated from the model architecture.

Predictions generated by neural networks are uncalibrated. It's questionable to directly measure the confidence of the models by their raw prediction values as in Section 4.


#########################################################################

---

> ### Author Response · Authors · 2020-11-23
> **Response to Reviewer 3**
>
> We thank the reviewer for their comments and constructive feedback.
>
> **[Paper lacks novelty]**  Please see our reply to all reviewers above.   The ability to train and classify on 5% image subsets with comparable performance to full images, and the consequences of this result that we outline are novel.   We have detailed how overinterpretation differs from previous work in this response, and provide seven references.
>
> **[Investigate the cause of this behavior]**  We do point out that this is a property of the dataset and demonstrate overinterpretation by different architectures.  Our results show that the datasets contain inherent statistical shortcuts that classifiers optimized for accuracy can learn to exploit, instead of learning more complex semantic relationships between the image pixels and the label.
>
> **[Predictions generated by neural networks are uncalibrated]**  We thank the reviewer for pointing this out, and we will revise the paper to clarify that these predictions are not explicitly calibrated and the consequential limitations of this lack of calibration.  The confidence scores we utilize do identify a statistically significant increase in SIS size for correctly vs. incorrectly classified images at multiple sampled confidence thresholds (Figure 4, Figure S3).

---

### Official Review · AnonReviewer1 · 2020-10-28
**Interesting experiments but unrelated claims**

**Rating:** 3
**Confidence:** 5

**Review:**

The work utilizes the SIS (a local feature-importance method) to empirically prove that on existing benchmark datasets, the trained convnets are capable of making decisions based on a very small subset of pixels that are meaningless to the human observer but are nonetheless strong signals. Interestingly, unlike common belief, the same phenomenon is observed in adversarially trained models. The main problem with the work is the discrepancy between claims and results.
Pros:
* The observed phenomenon is interesting and the work is novel
* The experimental setup is comprehensive.
* The writing is decent.

Drawbacks
* The main drawback with the work is its claim:
"that the highly sparse subsets found via backward selection offer a valid predictive signal in the CIFAR-10 benchmark".
"these sparse pixel-subsets are underlying statistical signals that suffice to accurately generalize from the benchmark training data to the benchmark test data".

I'm not so sure:

1- First, the fact that the observed phenomenon is very model-dependent (other models get low accuracy). This is against the assumption that there generally there exist sparse features that correlate with the class label. It might simply be a nuance of each architecture. One important modification that seems necessary is to create the training set using one architecture and then test the hypothesis using a different one. Just think about a simple scenario: In each architecture, the location of the interpretation mask tends to correlate with the class labels (important features of boat appear at bottom and dog appear at the top). In this case, the observed phenomenon is totally expected but not surprising. It seems like all the experiments on the sparse features are trained "and" tested with the "corresponding" dataset. I personally cannot be sure that the observed results are not simply indicative of the correlation between class labels and the shape of the SIS masks. One simple way of answering this question is how good a model trained on the sparse features is on clean images. If the answer is yes, then one can claim that there is enough signal that the chosen sparse subsets highly correlate with the label. Note that the reverse experiment (high accuracy of models trained on clean data when predicting sparse images) is not enough as in this experiment, the model itself is used to generate the sparse subset of features (which again means that it's a model-specific mask).

* The tone of the work suggests that this behavior means that existing models will be fragile for out of distribution data. While that might be true, this paper's observations as mentioned above, do not provide enough evidence. The work should either show such OOD samples or create a set of sparse images of CIFAR10 that are classified with high accuracy using any CNN architecture trained on clean images.

Questions and notes:
* It seems like choosing 5% of images randomly captures a large amount of signal. This is a very interesting observation.
* The interpretation being 5-10% of the images is not necessarily an indicator of poor behavior. Although the shown examples are indicative, they are hand-selected. I would like to see the average total variation of masks reported in order to have an idea about how scattered the important pixels are one average
* The work seems very related to adversarial examples are features, not bugs work, the towards automatic concept-explanations work, and ImageNet-trained CNNs are biased towards texture; increasing shape bias improves accuracy and robustness. It would be good to explain the relation of this work to each in detail.

---

> ### Author Response · Authors · 2020-11-23
> **Response to Reviewer 1**
>
> We thank the reviewer for the constructive feedback and interest in our work.
>
> **[Relationship to previous work]**  Please see our reply to all reviewers above.  We have detailed how overinterpretation differs from these previous references and other known flaws of deep image classifiers.
>
> **[Train models of 5% subsets from other architectures]**  We agree with the suggestion to create 5% pixel-subsets of CIFAR-10 images using one architecture and train other architectures on these subsets.  We ran this experiment using all architectures and found that models trained and tested on 5% pixel-subsets drawn from a different architecture perform as well as a new model of the original type on those subsets.  We have included complete results of these experiments in Table S2 of our revised paper.
>
> **[Train models on 5% subsets and test on clean images]**  We considered the suggested experiment to train a model on 5% pixel-subsets and test on fully unmasked images.  However, full images are highly out-of-distribution for a model trained on images with only 5% unmasked pixel-subsets and hence such a model cannot properly generalize to fully unmasked images.  Further, the model trained on 5% images does not learn to rely on the same features as the model trained on full images (it is a new model trained via nonconvex optimization on different data and therefore can learn an entirely  different function).  We also note our previous finding that even replicates of the same architecture (trained with different random initializations on the same dataset of full images) cannot accurately classify 5% pixel-subsets from other replicates (Section 4.1).  This result suggests there are multiple statistical patterns that a flexible model might learn to rely on, and thus CIFAR-10 image classification is a severely underdetermined problem.
>
> **[Correlation between mask location and labels]**  The reviewer raises a concern about a scenario in which the shape of the SIS masks is correlated with the class labels.  In our work, we use the SIS method to identify features that a model’s decision-making relies on and assume that SIS has accurately identified these features (see the original SIS paper by Carter et al. for an analysis of this capability).  To verify this assumption, we check that the resulting 5% pixel-subsets are indeed predictive signals to the model by confirming that they are just as accurately classified by the model as the original images.  The purpose of our experiment training a new model on these 5% subsets is *not* to prove that the original model relied on these features, but rather to show that these 5% features are indeed valid statistical signals in the CIFAR-10 benchmark (that is, these features generalize from the train to test set).  We indeed find that models trained on 5% pixel-subsets can generalize to the corresponding 5% pixel-subsets of test images.  Thus we believe our claim that the sparse pixel-subsets of CIFAR-10 and ImageNet images that we have identified (that are semantically meaningless to humans) offer a statistically valid predictive signal for these benchmarks is supported by the experiments in Section 4 of our paper.  In short, this paper shows that: (1) CNN models can learn to make decisions based on such tiny pixel subsets, (2) such non-salient pixel subsets are valid statistical patterns that can enable a new model to generalize to holdout data. We believe these findings are novel and important discoveries to share with the scientific community.
>
> **[Average total variation of masks]**  The reviewer also asks about average total variation of masks, which we included in Figure 2 for CIFAR-10 and ImageNet.  Figure 2a shows the pixel locations of all pixel-subset masks drawn from all 10000 CIFAR-10 test images by each architecture.  Figure 2b shows SIS pixel locations for a sample of 1000 ImageNet validation images (selected at random).
>
> **[Random 5% of images captures a large amount of signal]**  We agree that this is an interesting finding, and further reinforces the caution that should be used in interpreting results from these popular benchmark datasets.  It is another indication of the degeneracy of these datasets and is a troubling finding of the paper that random selections of downsampled pixels actually enable classification 5x better than random guessing.

---

### Official Review · AnonReviewer2 · 2020-11-02
**Interesting use of interpretability tools to highlight data biases/statistical artifacts.**

**Rating:** 6
**Confidence:** 3

**Review:**

This work reports the problem of image classification datasets (CIFAR-10 and ImageNet) which contains statistical patterns present in both training and tests that can be leveraged by neural networks to achieve high accuracy, but would not be discerned as salient features by humans. Using Sufficient Input Subsets (SIS), they show that retaining the smallest SIS to keep a confidence of 99% leads to spare sets of about 5% of the original pixels and that these subsets of pixels are not salient features for humans. Most importantly they show that training NNs on these SIS from a previously trained network achieves similar results.

===============================

Pros:

1. The paper targets a very important subject. Benchmarks are a fundamental part of the progress in machine learning but overreliance on a single metric can be problematic.
2. The results on models trained on tiny SIS but achieving nevertheless high accuracies are surprising and a sign that statistical artifacts (or sampling bias) are present in both training and test sets (the authors name it ‘valid statistical patterns’).
3. The experiments with models ensembles and input dropout show that the issue of overinterpretation can be alleviated to some degree.

Cons:

1. I am strongly confident that the pathology should not be blamed on the models but rather on the data. I am confident humans trained on the tiny SIS can learn to classify the examples with much greater accuracy than 20%. (More on this in Additional observations below)
2. Related to the last point, there is no mention in the paper (or I missed it) that all datasets studied in the original paper where SIS is proposed (Carter et al. 2019) do not lead to overinterpretation issues.
3. SIS and Batched SIS are not clearly presented and defined in the main text.

===============================

Reasons for score:

I would vote for a weak accept. The message of the paper has important implications, we cannot use interpretation tools if models use incomprehensible features that are statistical artifacts, or rather quoting the authors ‘interpretability method that faithfully describes the model should output these nonsensical rationales’. However, the presentation of the paper should be improved, for instance there lacks explanation of SIS and Batched SIS in the main paper to help the reader follow. Also, the experiments about methods to alleviate overinterpretation should be presented with results of human scores to better convey the suspected decrease of overinterpretation.

===============================

Additional observations

I believe the observations are mainly pathologies of the data rather than the models. If such statistical patterns exist in the data that allows generalization, then a model learning such features is not pathological but rather well adapted to its task. Methods to alleviate such issues by biasing models to rely on larger subsets of the input is to me only a way of alleviating the data’s sampling bias. Cormier et al 2019 do not indeed report such pathologies, most probably because the data they studied did not have this issue.

Humans are not trained only on CIFAR-10 to know what a frog or a car is. As authors say in section 4.2, there are statistical artifacts in the dataset that the humans do not know. I am fairly confident a human could be trained on the sparse versions and classify well the test example afterwards. The labels should also be changed to meaningless ones (ex: A, B, C, D, …) so that humans are learning from scratch like the CNNs.


I first thought the following to quotes to be contradictory:

‘We also find SIS subsets confidently classified by one model do not transfer to other models. For instance, 5% pixel-subsets derived from CIFAR-10 test images using one ResNet18 model (which classifies them with 94.8% accuracy) are only classified with 25.8%, 29.2%, and 27.5% by another ResNet18 replicate, ResNet20, and VGG16 models , respectively, suggesting there exist many different statistical patterns that a flexible model might learn to rely on [...].’

‘We find models trained solely on these pixel-subsets can classify corresponding test image pixel-subsets with minimal accuracy loss compared to models trained on full images. [...] This result suggests that the highly sparse subsets found via backward selection offer a valid predictive signal in the CIFAR-10 benchmark explointed by models to attain high test accuracy.’

After rereading many times I realized the first quote was about using SIS for one replicate on test set and compute test accuracy of another replicate using the same SIS, while the second quote was about training another replicate on the SIS and evaluating it on them. I think this could be made more clear in the text.

===============================

Questions

What is the precise threshold for the SIS? Is it always 99% confidence? But not all examples are predicted with 99% confidence isn’t? Is it 99% of f(original x)?

There must be a drawback with the Batched Gradient SIS algorithm, like lesser accuracy. Do the authors discuss it in the appendix? I have not found any discussion on that matter.

===============================

Typos

Page 3: [...] for the model to the same -> for the model to make the same
             [...] a gradient-based to find -> a gradient-based method to find?

===============================

Post-Rebuttal

I thank the authors for their detailed answers.  After reading the other reviews and the author's rebuttal, I maintain my rating of 6 for the paper. My concerns on the description of the SIS methods and results on the proposed mitigation are not addressed.

I am not convinced as R1 and R4 that training on the SIS and testing on the full image is the correct way of testing if SIS is sufficient for the model's predictions. If we would present SIS images with unrelatable labels (A instead of Cat, B instead of Dog, C instead of Boat, etc) to humans and ask them to learn the mappings, I am confident they could achieve good results. As pointed out by other reviewers we can see some patterns in the SIS. Showing a full image afterwards and asking to predict (A, B, C, ...) would be quite difficult however. It's easy to infer the pattern from the full image, but the other way around is more difficult. To me the most important is that a given model architecture can be trained on the SIS of another trained model (with different random initializations) and still be able to learn and generalize. That alone shows in my opinion that the dataset contains undesirable statistical artifacts shared by training and test sets, and as the authors says in the paper '‘interpretability method that faithfully describes the model should output these nonsensical rationales’.

I believe R4 makes a valuable point when saying '[...] I think it might be more useful to look at the mean SIS size of those that are wrongly classified by humans'. This seems to be a better way of gauging what threshold should be used for the size.

---

> ### Author Response · Authors · 2020-11-23
> **Response to Reviewer 2**
>
> We thank the reviewer for the feedback and interest in our work.   We will improve the writing for clarity where indicated by the reviewer.
>
> **[What is the precise threshold for the SIS?]**   In Figure 1 (SIS on CIFAR-10 images), all examples shown were computed using a threshold of 0.99 (99%+ confidence on the original image and on the SIS shown).   In Figure 2b and 3 (SIS on ImageNet images), a threshold of 0.9 was used.  Where we identify 5% pixel-subsets for CIFAR-10 (10% for ImageNet), the backward selection procedure of SIS is run on all images using the predicted class and the final 5% (10%) of pixels retained.   Note that the threshold is only needed to find a SIS after running backward selection, and for the purposes of selecting fixed-size subsets of images, no threshold is needed, and hence we are able to run this procedure on all images (regardless of initial confidence).   We verify that the classifiers can still predict as accurately on these 5% subsets of CIFAR-10 images (Table 1).
>
> **[Drawbacks to Batched Gradient SIS algorithm]**   We note that the Batched Gradient SIS algorithm is always completely accurate in that the SISs produced are guaranteed to be sufficient at a given threshold.  Thus every Batched Gradient SIS  is guaranteed to have confidence over the specified threshold.   However, Batched Gradient SIS is a gradient-based approximation to SIS, and thus a Batched Gradient SIS may be larger than those identified by the original SIS method.  The exact amount an SIS may be larger is dependent on the particular dataset and model.  However, for the purposes of this paper, larger SIS sizes are conservative as the identified subsets on ImageNet images are still spurious.  Figure S10 shows the confidence on remaining (unmasked) pixels at each step of Batched Gradient BackSelect to show that the majority of pixels can be removed while still retaining high confidence toward the predicted class.
>
> **[Overinterpretation is mainly a pathology of the data rather than models]**   We agree.   As we point out, this is a property of the data.  We find popular image datasets contain these artifacts, and the resulting overinterpretation may be difficult to overcome with ML methods alone.

---

### Author Response · Authors · 2020-11-23
**Response to all reviewers**

We thank the reviewers for their interest in our work and for their constructive feedback.   In particular, we are glad to find that reviewers consider the observed phenomenon interesting and the work novel (R1), our paper to have important implications (R2), the pathology realistic and shared by mainstream CNN architectures (R3), and our experiments extensive (R4).   In the replies to each review below, we address questions raised by each reviewer.

While some reviewers found our work novel, other reviewers needed clarification on how overinterpretation differs from previous work.  While existing work has demonstrated numerous distinct flaws in deep image classifiers (adversarial examples, relying on spurious signals like texture/background, etc.), our paper demonstrates a *new* distinct flaw, overinterpretation, previously undocumented in the literature.  It is important to inform researchers about this new flaw, and thus we comprehensively contrast overinterpretation against other known flaws below:

- Ghorbani et al. (2019) introduce principles and methods for human-understandable concept-based explanations of ML models.  In contrast, overinterpretation differs because the features we identify are semantically meaningless to humans, stem from single images, and are not aggregated into interpretable concepts.  The existence of such subsets stemming from unmodified subsets of images suggests degeneracies in the underlying benchmark datasets and failures of modern CNN models to rely on more robust and interpretable signals in training datasets.

- Geirhos et al. (2019) discover that ImageNet-trained CNNs are biased towards recognizing textures rather than shapes.  Overinterpretation reveals a different failure mode in which ImageNet and CIFAR-10-trained CNNs latch onto spurious, semantically meaningless features comprising highly sparse pixel-subsets.  Our discovery that CNNs rely on these non-salient features suggests they may not generalize well to out-of-distribution data or be dangerous to deploy in real-world scenarios.

- Geirhos et al. (2018) discover that DNNs trained on distorted images fail to generalize as well as human observers when trained under image distortions.  In contrast, overinterpretation reveals a different failure mode of DNNs, whereby models latch onto spurious but statistically valid sets of features in undistorted images.  This phenomenon can limit the ability of a DNN to generalize to real-world data even when trained on natural images.

- Rosenfeld et al. (2018) show the fragility of image classifiers when objects from one image are transplanted in another image.  In contrast, overinterpretation differs because we demonstrate that highly sparse, unmodified subsets of pixels in images suffice for image classifiers to make the same predictions as on the full images.  Our analysis showing that such signals generalize to the test distribution suggests that these signals arise from degenerate signals in popular benchmarks, and thus models trained on these datasets may fail to generalize to real-world data.

- Brendel et al. (2019) show that CNNs trained on natural ImageNet images may rely on local features and, unlike humans, are able to classify texturized images.  They suggest that ImageNet alone is insufficient to force DNNs to rely on more causal representations.  In contrast, our work demonstrates another source of degeneracy of popular image datasets, where sparse, unmodified subsets of training images that are meaningless to humans can enable a model to generalize to test data.  Our work provides one explanation for why ImageNet-trained models may struggle to generalize to out-of-distribution data.

- Lapuschkin et al. (2019) demonstrate that DNNs can learn to rely on spurious signals in datasets, including source tags and artificial padding, but which are still human-interpretable.  In contrast, the patterns we identify are minimal collections of pixels in images that are semantically meaningless to humans (they do not comprise human-interpretable parts of images).  Further, we demonstrate that these non-salient pixel subsets are valid statistical patterns that can enable a new model to generalize to holdout data.

- The susceptibility of DNNs to adversarial examples or synthetic images has been widely studied (e.g., Goodfellow et al. (2014), Nguyen et al. (2015), Madry et al. (2018); Ilyas et al. (2019)).  However, these adversarial examples synthesize artificial images or modify real images with auxiliary information.   In contrast, we demonstrate overinterpretation of unmodified subsets of actual training images, indicating the patterns are already present in the original dataset, and further show that such signals in training data actually generalize to the test distribution.   We also demonstrate that adversarially robust models also suffer from overinterpretation.

We will revise Section 2 to more clearly delineate how overinterpretation differs from previous work.

---

> ### Author Response · Authors · 2020-11-23
> **References for related work**
>
> References
>
> Brendel, W., & Bethge, M. (2019). Approximating cnns with bag-of-local-features models works surprisingly well on imagenet. arXiv preprint arXiv:1904.00760.
>
> Geirhos, R., Temme, C. R., Rauber, J., Schütt, H. H., Bethge, M., & Wichmann, F. A. (2018). Generalisation in humans and deep neural networks. In Advances in neural information processing systems (pp. 7538-7550).
>
> Geirhos, R., Rubisch, P., Michaelis, C., Bethge, M., Wichmann, F. A., & Brendel, W. (2019). ImageNet-trained CNNs are biased towards texture; increasing shape bias improves accuracy and robustness. In International Conference on Learning Representations.
>
> Goodfellow, I. J., Shlens, J., & Szegedy, C. (2014). Explaining and harnessing adversarial examples. arXiv preprint arXiv:1412.6572.
>
> Ghorbani, A., Wexler, J., Zou, J. Y., & Kim, B. (2019). Towards automatic concept-based explanations. In Advances in Neural Information Processing Systems (pp. 9277-9286).
>
> Ilyas, A., Santurkar, S., Tsipras, D., Engstrom, L., Tran, B., & Madry, A. (2019). Adversarial examples are not bugs, they are features. In Advances in Neural Information Processing Systems (pp. 125-136).
>
> Lapuschkin, S., Wäldchen, S., Binder, A., Montavon, G., Samek, W., & Müller, K. R. (2019). Unmasking clever hans predictors and assessing what machines really learn. Nature communications, 10(1), 1-8.
>
> Madry, A., Makelov, A., Schmidt, L., Tsipras, D., & Vladu, A. (2018). Towards Deep Learning Models Resistant to Adversarial Attacks. In International Conference on Learning Representations.
>
> Nguyen, A., Yosinski, J., & Clune, J. (2015). Deep neural networks are easily fooled: High confidence predictions for unrecognizable images. In Proceedings of the IEEE conference on computer vision and pattern recognition (pp. 427-436).
>
> Rosenfeld, A., Zemel, R., & Tsotsos, J. K. (2018). The elephant in the room. arXiv preprint arXiv:1808.03305.

---

### Decision · Program_Chairs · 2021-01-07
**Final Decision**

**Decision:**

Reject

**Comment:**

The reviewers generally feel that the phenomenon discovered in this paper is relevant and could be very important when considering interpretability. However, there are still a number of remaining concerns. The reviewers are not convinced by the human study - they feel there is structure in the SIS’s such that a human trained on these images with an abstract category (i.e., without being told their real-world label) could potentially successfully learn to classify them. There is also a concern that SIS is model-based, that is, the inductive biases of the model (shape, color, etc.) could be leaking information into the SIS image. Finally, there should be some stronger evidence that this represents a serious practical problem for the community. Are there instances where current interpretable approaches break down because of this phenomenon?

One suggestion to potentially strengthen the human experiment: you could try training a denoising autoencoder on the full images, removing 95% of the pixels at random. Then, given an SIS, use the denoising autoencoder to reconstruct the image and then provide that to a human subject. The question is: how much information about the image as a whole is preserved in the SIS (when combined with an appropriate inductive bias)?